# Comparing Wetland Ecosystems Service Provision under Different Management Approaches: Two Cases Study of Tianfu Wetland and Nansha Wetland in China

Yuchao Zhang [1,*], Steven Loiselle [2,3], Yimo Zhang [4], Qian Wang [4], Xia Sun [4], Minqi Hu [1,5], Qiao Chu [1,5] and Yuanyuan Jing [1,5]

1   Key Laboratory of Watershed Geographic Sciences, Nanjing Institute of Geography and Limnology, Chinese Academy of Sciences, Nanjing 210008, China; huminqi16@mails.ucas.ac.cn (M.H.); chuqiao17@mails.ucas.ac.cn (Q.C.); jingyuanyuan18@mails.ucas.ac.cn (Y.J.)
2   Dipartimento di Biotecnologie, Chimica e Farmacia, University of Siena, CSGI, Via Aldo Moro 2, 53100 Siena, Italy; sloiselle@earthwatch.org.uk
3   Earthwatch Europe, 256 Banbury Road, Oxford OX2 7DE, UK
4   WWF China, Beijing 100037, China; ymzhang@wwfchina.org (Y.Z.); wangqian@wwfchina.org (Q.W.); xsun@wwfchina.org (X.S.)
5   University of Chinese Academy of Sciences, Beijing 100049, China
*   Correspondence: yczhang@niglas.ac.cn

**Abstract:** The largest blue-green infrastructures in industrialized, urbanized and developed regions in China are often multiuse wetlands, located just outside growing urban centers. These areas have multiple development pressures while providing environmental, economic, and social benefits to the local and regional populations. Given the limited information available about the tradeoffs in ecosystem services with respect to competing wetland uses, wetland managers and provincial decision makers face challenges in regulating the use of these important landscapes. In the present study, measurements made by citizen scientists were used to support a comparative study of water quality and wetland functions in two large multiuse wetlands, comparing areas of natural wetland vegetation, tourism-based wetland management and wetland agriculture. The study sites, the Nansha and Tianfu wetlands, are located in two of the most urbanized areas of China: the lower Yangtze River and Pearl River catchments, respectively. Our results indicated that the capacity of wetlands to mitigate water quality is closely related to the quality of the surrounding waters and hydrological conditions. Agricultural areas in both wetlands provided the lowest sediment and nutrient retention. The results show that the delivery of supporting ecosystem services is strongly influenced by the location and use of the wetland. Furthermore, we show that citizen scientist-acquired data can provide fundamental information on quantifying these ecosystem services, providing needed information to wetland park managers and provincial wetland administrators.

**Keywords:** STP activity; wetland ecosystem services; water quality improvement

## 1. Introduction

Blue-green infrastructure approaches have been adopted in many urban areas to combat climate change and environmental degradation (https://theconstructor.org/sustainability/blue-green-infrastructure/555236/, (accessed on 3 August 2021)). Blue-green infrastructure is most often associated with a network of natural and semi-natural areas, offering environmental, economic, and social benefits to communities [1]. Since they have been shown to play an important role in maintaining water supply and water quality control, the value of blue-green infrastructure has gained greater attention in the last two decades [2]. Wetlands are the most common natural blue-green infrastructure [1] and provide well-defined ecosystem services [3,4], including (1) food provision, (2) flood prevention/mitigation, (3) improvement of water quality and microclimate, (4) conservation of biodiversity, and

(5) social/cultural functions. There have been many studies exploring the mechanisms related to the improvement of surface runoff water quality in wetlands, such as nitrate loss by denitrification [5,6] and phosphorus removal by soil absorption [7,8].

However, many wetlands have multiple and competing land uses and development pressures. Few studies have compared the relative benefits on water quality mitigation related to different wetland uses. Such information is needed to allow decision makers to create a more integrated wetland management policy.

In the present study, two large multiuse wetlands were chosen to compare the effects of wetland utilization on its capacity to mitigate poor water quality in incoming waters. The Tianfu wetland in the Yangtze Delta and the Nansha wetland in the Pearl Delta are major blue-green infrastructures in areas under high development. The possibility to examine different types of wetland management, in relation to a key ecosystem service, is a unique and important opportunity to provide provincial and national decision makers with better tools to manage these important landscapes. Furthermore, there is a clear need for robust scientific data that allow such a comparison in the multiuse wetlands that dominate eastern and southern China.

The Sustainability Training Program (STP) is a global program initiated by the HSBC Group, with Earthwatch responsible for designing and leading the implementation. This program aims to raise awareness of sustainability among HSBC staff, incentivize them to take action to support sustainable business practices and encourage everyday action to support sustainable corporate growth. In 2018 & 2019, thirteen measurement events were carried out in Nansha and Tianfu wetlands, gathering a total of 2664 monitoring datasets, by 251 HSBC citizen scientists trained by NIGLAS researchers.

By comparing data gathered by trained citizen scientists in different areas of land use in the Tianfu and Nansha wetlands over 2018 and 2019, our study focused on (1) examining the temporal and spatial variations in water quality across different wetland areas, (2) comparing the water quality differences between upstream and downstream in different wetlands; and (3) evaluating effects of different land uses on wetland water quality mitigation services, to provide support for decision makers on land use and water management in multiuse blue-green infrastructures in developing areas.

## 2. Methodology

### 2.1. Study Area

The Tianfu wetland in the Yangtze Delta and Nansha wetland in the Pearl River Delta (Figure 1) provide key services that make these wetlands the cradles of prosperity in these rapidly growing areas. In recent decades, these two areas have also grown to two of the most industrialized, urbanized and developed regions in China. Meanwhile, the distance between nature and citizens in these cities has grown. The creation of wetland parks has allowed people in urban areas to experience nature, by combining hiking, birdwatching and interaction areas for parents and their children.

At the same time, agricultural utilization of wetlands has both traditional and economic value in China, and both wetlands have a large area dedicated to the production of regionally important crops. On the other hand, regional authorities have a growing understanding of the importance of conservation actions in these key ecosystems for biodiversity and other ecosystem services. Typical of many blue-green spaces, all three objectives are present in the Tianfu and Nansha wetlands, with a managed wetland park (disturbed area), a conservation area (natural wetland) and agricultural wetland competing for space. To explore differences between uses in wetland ecosystem services related to water quality, 6 fixed monitoring points in each wetland were identified, set at the upstream and downstream of the natural wetland, agricultural wetland and wetland park.

The Tianfu wetland is a typical paddy field wetland of 7.84 km$^2$ located in Huaqiao Town, Suzhou City, Jiangsu Province. It has a south monsoon climate typical of the north subtropical zone. The annual average temperature is 15.7 °C and precipitation exceeds

1000 mm. The agricultural wetland area has an annual double crop rotation system, with wheat, rapeseed and rice being the most popular crops.

**Figure 1.** Locations of Tianfu wetland and Nansha wetland.

Nansha wetland is a typical coastal estuary wetland located in Nansha District, Guangzhou City, Guangdong Province, with a total area of 6.85 km$^2$. The wetland is the focus of two major projects of restoration and utilization. The first phase of the Nansha wetland project (2.27 km$^2$) has been recently completed, creating a core area of ecological protection used for scientific education and research. The second phase is focused on ecotourism and recreation and is still in the beginning stage of development. The main ecological communities in Nansha wetland are mainly mangroves and reeds. It has a subtropical marine monsoon climate with an annual average temperature is 21.9 °C and precipitation above 1500 mm. Crops include bananas, sugarcane, rice and lotus root. Fish, shrimp and other aquatic products are also harvested within the wetland.

### 2.2. Citizen Science Methods and Data

2.2.1. Citizen Scientist Training

Participants were trained in assessment methods and made measurements in multiple sites in each wetland as citizen scientists (guided and supported by professional researchers) over a 2-year period. In each wetland, they observed optical, chemical and biological characteristics of the water and vegetation in specific sites upstream and downstream of wetlands that are managed for conservation purposes, tourism purposes and agricultural purposes.

In 2018 & 2019, thirteen training and monitoring activities were carried out in Nansha wetland and Tianfu wetland (Figure 1), respectively. This was complemented by additional sampling collection by NIGLAS researchers (Table A1). For each water quality monitoring

activity, three types of wetlands and six monitoring points in the upstream and downstream were included. A total of 2664 datasets were collected by 251 HSBC citizen scientists and NIGLAS researchers.

### 2.2.2. Water Quality Indicators

Citizen scientists collected ecosystem data, hydrological data and water quality data by field observation and simple water quality measurement kits at each monitoring point, while NIGLAS researchers collected water samples for laboratory monitoring (Table 1). This allowed for a temporal account of main water quality parameters, as well as the response of water quality parameters to different wetland types and spatial changes of upstream and downstream.

**Table 1.** Summary of sampling indicators. Full explanation of parameter acronyms is in Methods Section 2.4.

| Type | Sampling Way | Sampling Indicators |
|---|---|---|
| Ecosystem | In-site | Land Use, Vegetation Community Structure, Vegetation Density |
| Hydrology | In-site | Flow Velocity, Water Level |
| Water Quality | In-site | Secchi Disc, Water Color, Turbidity, $NO_3^--N$, $PO_4^{3-}-P$ |
| | Lab | TSM, ISM, OSM, Chl*a*, DOC, TN, TP, Salinity |

Integrated measurements from upstream and downstream sites of a natural wetland area, upstream and downstream of a disturbed tourism wetland, and upstream and downstream of an agricultural wetland were made. For each wetland area, three study sites were selected upstream of the wetland and three study sites were selected below the wetland (Figure 1).

### 2.2.3. In-Site Qualitative Measurements

Qualitative measurements of wetland vegetation were obtained by citizen scientists to characterize differences between the sites with respect to wetland management. Participants recorded information about:

- Vegetation density (31.6 cm × 31.6 cm square quadrant, pin quadrant method or stem count), repeated 3 times;
- Vegetation community (presence or absence of visible invasive species within a 10 m distance from sampling site);
- Surrounding land use (within 50 m from study site);
- Flow direction, which was decided on site.

### 2.2.4. In-Situ Quantitative Measurements

Quantitative measurements were made of the wetland water quality and water characteristics using state of the art optical instrumentation:

- The estimate of Secchi depth, using a standardized Secchi device (tube or disk for boat samples). Turbidity was assessed by a standardized Secchi tube from 12 and 240 Nephelometric Turbidity Units (NTU) [9,10];
- An estimate of water color components using a Forel-based color chart and comparing the color of the particulate matter (syringe filter color) after filtering 20 mL of lake water. The filtrate (dissolved fraction) was measured using a portable colorimeter at 440 nm (1 colorimeter per team). These measurements were used to identify the types and concentration of particulate matter (filter color) and dissolved matter (filtrate). Three nearby measurements in each site (within 10 m from each other) were made to compare spatial heterogeneity;
- An estimate of dissolved nutrients (nitrate and phosphate) from the filtrate of the samples taken above using standard colorimetric methods [11]. Nitrate and phosphates were tested in enclosed reagent tubes and compared with corresponding colors [12].

## 2.3. Land Cover Classification and Validation Methods

Data from the Sentinel-2 satellite system were used to gather information about vegetation, soil and water conditions. In this study, 2 Sentinel-2A/MSI Level-2A products with limited clouds and aerosol conditions were selected to explore land cover use of Tianfu wetland and Nansha wetland in 2019.

The classification approach was based on Support Vector Machine (SVM) methods, using training data from Google Earth. This information was combined with the time series of vegetation surveys and land use maps of 2015 to determine wetland water sampling sites. About 200 points were chosen in the training sample set. The accuracy of land use maps for these two wetlands was evaluated using a confusion matrix based on 100 field samples in 2018 and 2019 (Figure 1). The errors of commission and omission, the overall accuracy and Kappa coefficient for classification results were determined.

## 2.4. Laboratory Analysis Method

Samples (2 L) of water were obtained from each site (one per site) for analysis in the NIGLAS laboratory of total suspended matter (TSM), inorganic suspended matter (ISM), organic suspended matter (OSM), chlorophyll *a* (Chl*a*), dissolved organic carbon (DOC), total nitrogen (TN), total phosphorus (TP) and salinity.

A part of these samples (50–200 mL) was filtered through pre-combusted and pre-weighed Whatman GF/F filters, and subsequently dried at 105 °C for 4 h to determine SPM concentrations gravimetrically. The SPM was further differentiated into suspended inorganic matter (SPIM) and suspended organic matter (SPOM) by combusting the organic matter from the filters at 450 °C for 4 h and reweighing the filters [13,14]. A Whatman GF/F glass fiber filter was used to filter another part of the water samples and subsequently soaked in 90% acetone to extract the pigments, which was measured at 630 (A630), 645 (A645), 663 (A663), and 750 nm (A750) using a UV2600 spectrophotometer (Shimadzu, Japan) [15,16]. TN and TP concentration was determined by spectrophotometry after digestion with alkaline potassium persulfate [17]. The precise measure of salinity was provided by full ionic analysis followed by summation of all ion concentrations [18].

## 2.5. Data Analysis Methods

The Spearman rank correlation coefficient was used, as the data were not normally distributed [19], to obtain the relationship of water quality and different wetland types. T-tests (Student's *t*-test) were used to compare datasets before and downstream of the wetland types and compare wetland types. Variance in measurements between upstream and downstream were also evaluated.

## 2.6. Meteorological Data

Daily data (air temperature, wind direction, wind speed, precipitation, sunshine hours etc.) were obtained for the nearest national weather stations (i.e., Kunshan and Guangzhou Stations) from 2018 to 2019 from the China Meteorological Data Sharing Service System (http://cdc.cma.gov.cn/home.do, (accessed on 31 January 2021)).

## 3. Results

### 3.1. Land Use of Tianfu and Nansha Wetlands and Accuracy Evaluation

The confusion matrix and the errors of commission and omission for wetland classification results in 2019 had an overall accuracy for classification of 83.27% and a Kappa coefficient of 0.828. In total, the area of Tianfu wetland was determined to be 7.84 km$^2$, including water area of 0.88 km$^2$, grassland/shrub 1.89 km$^2$, cropland 3.59 km$^2$ and construction land 1.48 km$^2$. The total area of Nansha Wetland Park was determined to be 6.85 km$^2$, including water area of 1.31 km$^2$, grassland/shrub 2.21 km$^2$, cropland 3.19 km$^2$ and construction land 0.14 km$^2$. The area located between Fourteenth Chong River and Sixteenth Chong River was about 15.04 km$^2$, including water area 1.82 km$^2$, grassland/shrub 1.34 km$^2$, cropland 9.83 km$^2$ and construction land 2.05 km$^2$. The land use in these two wet-

lands shows significant heterogeneity in the uses of these different wetlands (Figure 2 & Table 2).

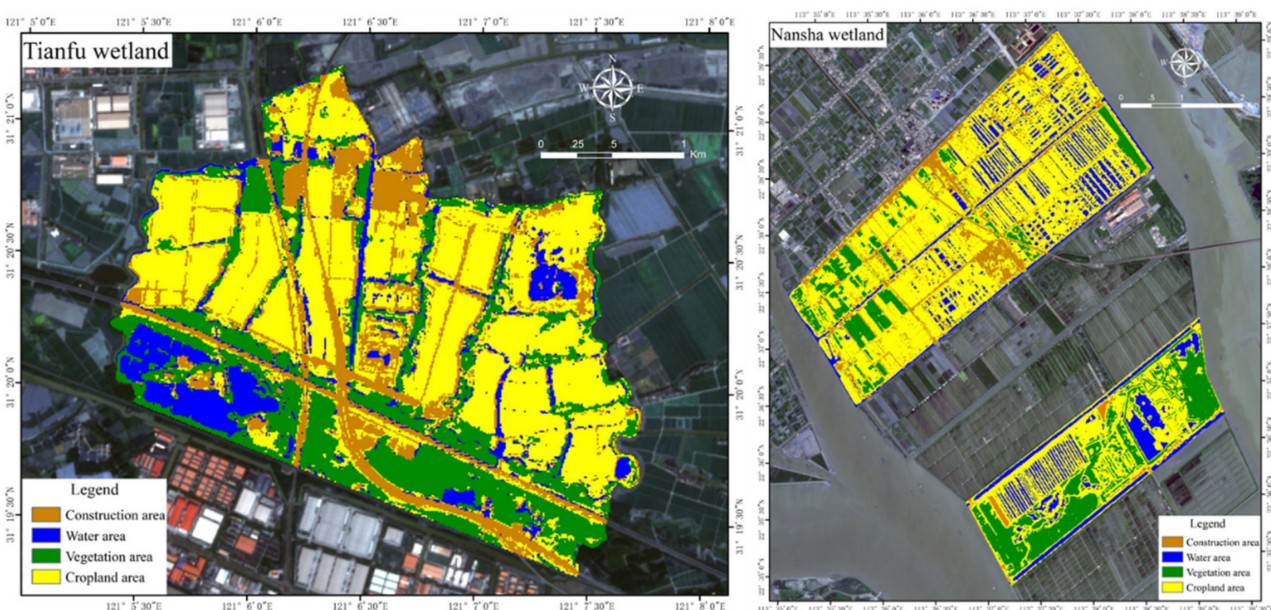

**Figure 2.** Land cover of Tianfu wetland and Nansha wetland in 2019.

**Table 2.** Land use composition of Tianfu wetland and Nansha wetland in 2019.

| Study Area | Total Area (km$^2$) | Water Area (km$^2$) | Grassland/Shrub (km$^2$) | Cropland (km$^2$) | Construction Land (km$^2$) |
|---|---|---|---|---|---|
| Tianfu wetland | 7.84 | 0.88 | 1.89 | 3.59 | 0.14 |
| Nansha wetland | 21.89 | 3.13 | 3.55 | 13.02 | 2.19 |

Wetland parks are very good places for people in these highly urbanized areas of China to experience nature, including hiking, birdwatching and interaction between parents and their children. To meet this tourism demand, these wetland parks have a high degree of land cover consisting of water area (>50%), construction and artificial shrubs. In the natural wetland area, ecosystem conservation requires the presence of natural wetland vegetation (>60%). In the agricultural wetland areas, cropland covers more than 80% of the total area.

*3.2. Water Quality Characteristics of Tianfu Wetland*

3.2.1. Summary Description of Water Quality Data

During the study, the Tianfu wetland was still undergoing alterations due to an influenced water circulation project, and hydrological flows were relatively static. The flow direction (Figure 2) was determined before alterations began. Water passing through both wetlands comes from flows originating in the larger basin.

In Figure 3, the water color for most sites ($\geq$83.1%) was determined to be yellow green or light brown, a water color level of No. 15 to 17. The water was totally clear and the average Secchi depth was about 90 cm. Rapid test kits showed that more than 85.6% and 79.8% of N-NO$_3$ and P-PO$_4$ concentrations were less than 0.5 mg/L and 0.05 mg/L, respectively.

Lab data (Table 3) showed that TSM concentrations were low, ranging between 2.67 to 38.4 mg/L. ISM and OSM were both about half of the TSM. The average concentrations of Chl*a* and DOC were about 16.8 μg/L and 13.4 ppm. TN and TP concentrations ranged between 0.08 to 1.81 mg/L and between 0.009 to 0.45 mg/L, respectively, which indicated the water of Tianfu wetland belonged to level II to level III surface water classification under the National Environmental Quality Standard for Surface Water of China (GB 3838-

2002) (SEPA 2002, Table A2). The salinity of all monitoring sites in Tianfu wetland was lower than 500 ppm, confirming its freshwater nature.

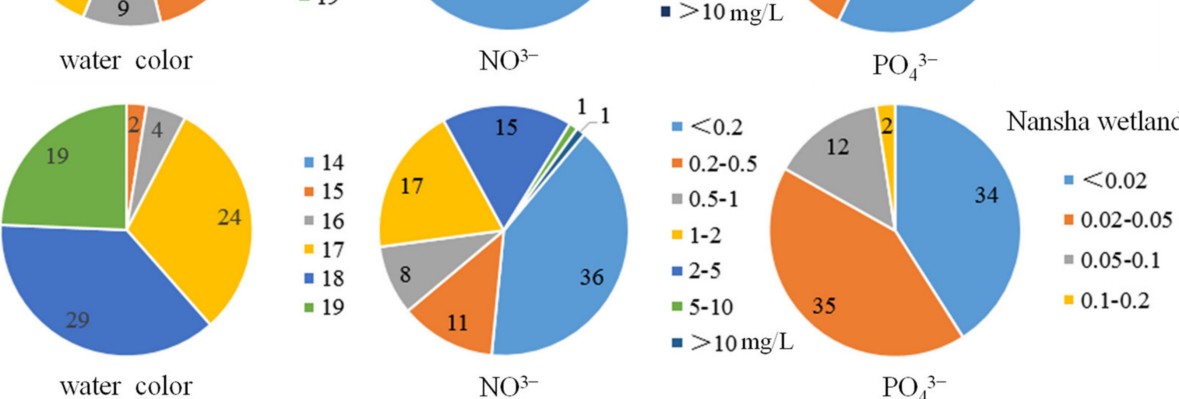

**Figure 3.** Summary description of water quality gathered by STP activities in the field by citizen scientists, including FUI water color, nitrate nitrogen (ppm) and phosphate phosphorus (ppm).

### 3.2.2. Temporal Dynamics of Water Quality in Tianfu Wetland

TSM, ISM and OSM in Tianfu were the highest in winter and summer and low in spring and autumn (shown in Figure 4). There was a strong variation in the proportion of organic matter and inorganic matter by season: inorganic matter dominated in winter, while organic matter dominated in summer, demonstrating the important contribution of phytoplankton growth to the wetland water quality. Variations of DOC were low in winter and high in summer, indicating the release of organic matter from the wetland. Although the Chl*a* confirmed the OSM data, an unexpected increase occurred between January to March of 2019. The change of TN in a year was similar to that of TSM, while TP was relatively stable with low value.

### 3.3. Water Quality Characteristics of Nansha Wetland

#### 3.3.1. Summary Description of Water Quality Data

The water in Nansha Wetland Park is mainly connected with the Eighteenth Chong River and Nineteenth Chong River through sluices which regularly switch according to the tide (Figure 1). Usually, the water flows from Eighteenth Chong River to Nineteenth Chong River when the sluices are open. The monitoring points of agricultural wetland are located in the upper and lower reaches of the Fifteenth Chong River, which flows from east to west.

**Table 3.** Characteristics of water quality analyzed in the laboratory.

| Wetland Type | Statistics | Water Quality Measurements | | | | | | | | |
|---|---|---|---|---|---|---|---|---|---|---|
| | | TSM (mg/L) | ISM (mg/L) | OSM (mg/L) | Chl*a* (μg/L) | DOC (ppm) | TN (mg/L) | TP (mg/L) | Secchi (cm) | Salinity (ppm) |
| Tianfu wetland | | | | | | | | | | |
| Wetland park | Samples/N | 30 | 30 | 30 | 30 | 30 | 28 | 28 | 30 | 22 |
| | Mean | 14.56 | 8.15 | 6.42 | 16.02 | 12.09 | 0.79 | 0.057 | 92.5 | 222.71 |
| | Max | 38.40 | 29.60 | 12.00 | 70.44 | 46.89 | 1.75 | 0.19 | 160.00 | 391.00 |
| | Min | 4.00 | 0 | 2.40 | 1.03 | 2.13 | 0.21 | 0.009 | 30.00 | 145.36 |
| | Stdv | 9.62 | 8.02 | 2.51 | 14.28 | 11.08 | 0.35 | 0.04 | 30.05 | 52.83 |
| Natural wetland | Samples/N | 30 | 30 | 30 | 30 | 30 | 28 | 28 | 30 | 22 |
| | Mean | 12.56 | 6.15 | 6.41 | 19.42 | 12.75 | 0.48 | 0.03 | 87 | 201.54 |
| | Max | 26.40 | 19.20 | 11.20 | 106.36 | 44.79 | 0.83 | 0.08 | 140.00 | 322.50 |
| | Min | 2.67 | 0 | 1.33 | 1.66 | 1.86 | 0.24 | 0.008 | 50.00 | 151.01 |
| | Stdv | 6.68 | 5.77 | 2.17 | 27.05 | 11.02 | 0.18 | 0.02 | 23.47 | 52.92 |
| Agricultural wetland | Samples/N | 30 | 30 | 30 | 30 | 30 | 28 | 28 | 30 | 22 |
| | Mean | 14.72 | 6.98 | 7.73 | 14.83 | 15.41 | 0.78 | 0.08 | 81.60 | 229.30 |
| | Max | 32.80 | 20.80 | 24.00 | 33.68 | 56.17 | 1.81 | 0.45 | 160.00 | 413.50 |
| | Min | 2.67 | 0 | 1.33 | 0.61 | 2.65 | 0.08 | 0.01 | 50.00 | 122.72 |
| | Stdv | 7.34 | 5.75 | 3.47 | 10.51 | 13.76 | 0.41 | 0.09 | 24.61 | 66.70 |
| Nansha wetland | | | | | | | | | | |
| Wetland park | Samples/N | 30 | 30 | 30 | 28 | 30 | 28 | 28 | 30 | 22 |
| | Mean | 33.11 | 19.65 | 13.46 | 30.94 | 7.84 | 1.28 | 0.07 | 42.17 | 1555.09 |
| | Max | 72.00 | 60.00 | 57.33 | 271.93 | 52.60 | 2.41 | 0.20 | 90.00 | 5350 |
| | Min | 11.20 | 4.00 | 4.00 | 0.96 | 1.32 | 0.37 | 0.004 | 20.00 | 170.35 |
| | Stdv | 19.24 | 16.09 | 10.63 | 49.74 | 11.93 | 0.55 | 0.05 | 15.96 | 1684.51 |
| Natural wetland | Samples/N | 29 | 29 | 29 | 27 | 29 | 27 | 27 | 18 | 21 |
| | Mean | 33.29 | 23.17 | 10.12 | 22.49 | 11.65 | 0.76 | 0.11 | 31.22 | 1548.29 |
| | Max | 115.20 | 98.40 | 17.60 | 149.27 | 68.15 | 1.80 | 1.46 | 52.00 | 4350 |
| | Min | 5.60 | 0.80 | 4.80 | 1.70 | 1.27 | 0.29 | 0.003 | 15.00 | 348.50 |
| | Stdv | 28.99 | 26.95 | 3.16 | 28.57 | 14.89 | 0.43 | 0.27 | 11.65 | 1146.54 |
| Agricultural wetland | Samples/N | 30 | 30 | 30 | 28 | 28 | 28 | 28 | 30 | 22 |
| | Mean | 30.72 | 21.37 | 9.35 | 18.31 | 6.80 | 1.60 | 0.10 | 33.03 | 504.13 |
| | Max | 65.60 | 55.20 | 25.33 | 66.01 | 50.05 | 2.71 | 0.40 | 50.00 | 1985 |
| | Min | 8.80 | 0 | 4.80 | 3.56 | 0.92 | 0.56 | 0.02 | 20.00 | 82.43 |
| | Stdv | 13.85 | 12.14 | 3.64 | 14.30 | 10.16 | 0.47 | 0.09 | 9.13 | 507.19 |

Compared to Tianfu, water quality in Nansha wetland was much poorer. Measurements showed more turbid waters, and the average Secchi depth was 35 cm. The water color for most sites (≥92.3%) was yellow or brown, the water color level of which is No. 17 to 19. Rapid test kits indicated that more than 57.3% of N-NO$_3$ concentrations ranged between 0.2 to 5 mg/L, which were much higher than those in Tianfu wetland. There was no significant difference in P-PO$_4$ concentrations between these two wetlands.

Together with lab data, these data indicated a higher eutrophication state in Nansha wetland with respect to Tianfu wetland. The TSM concentration of Nansha wetland ranged between 5.60~115.20 mg/L, and ISM accounted for more than 59.3%, of TSM. The range of Chl*a* and DOC concentrations were wide, with average concentrations of 23.91 μg/L and 8.76 ppm, respectively. TN and TP concentrations ranged between 0.29 to 2.71 mg/L and between 0.003 to 1.46 mg/L respectively. This puts Nansha wetland water into level IV, even level V, of national surface water area functions (GB 3838-2002) [20], typical of water in a higher eutrophication state. Salinity in Nansha fluctuated between 82.43 to 5350 ppm, showing the influence of the water exchange cycle between wetland and ocean.

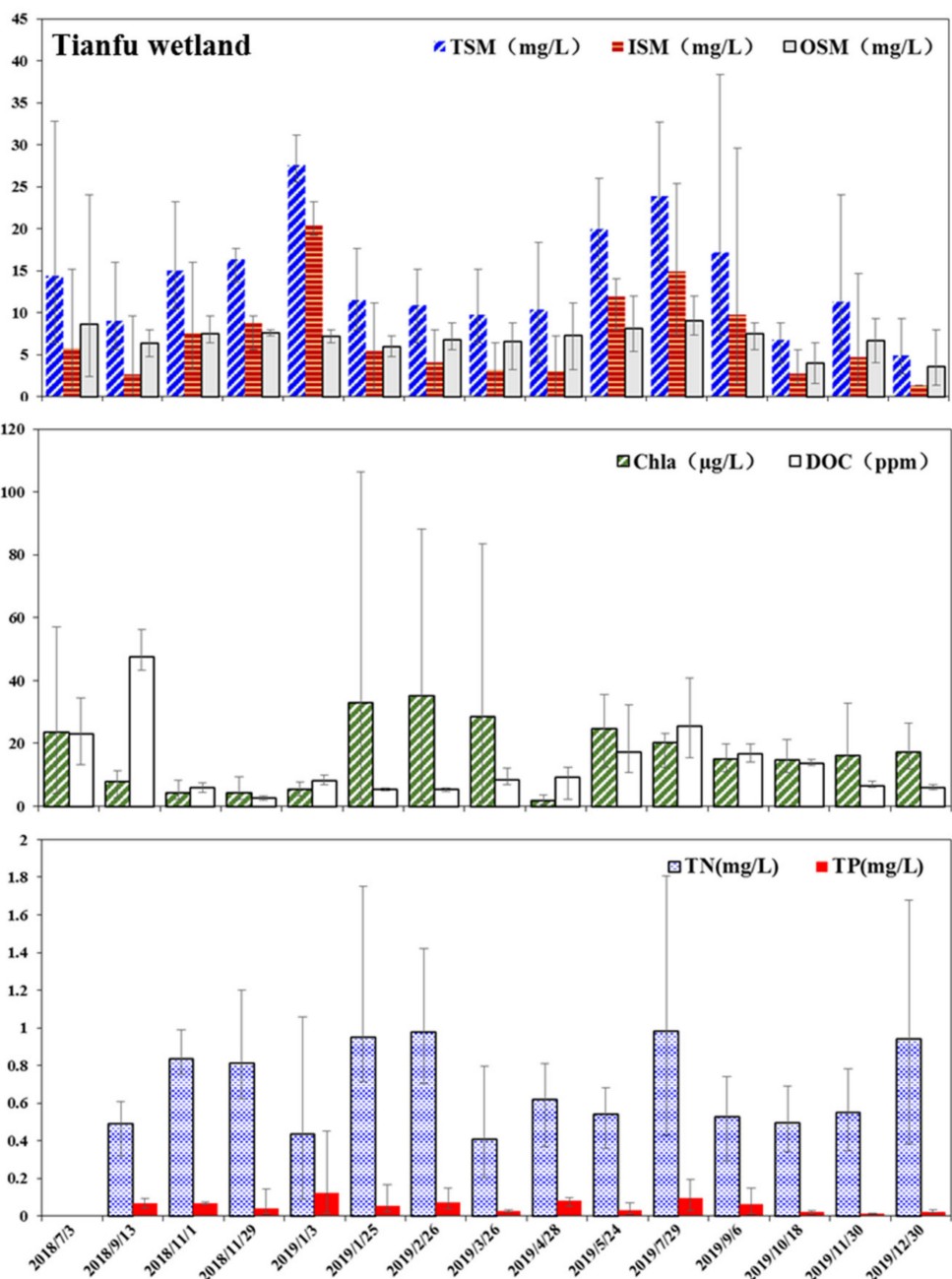

**Figure 4.** Temporal variations of water quality indicators in Tianfu wetland.

### 3.3.2. Temporal Dynamics of Water Quality in Nansha Wetland

The water in Nansha wetland was turbid all year around (shown in Figure 5). However, the concentrations of TSM, ISM and OSM in February to March were relatively low. Different from Tianfu wetland, ISM in Nansha wetland was dominant throughout the year. The concentrations of Chl*a* and DOC were higher in summer and lower in winter consistently. TN concentrations were high throughout the year, higher in winter and spring, and lower in summer and autumn. The rainy season in June was the shift point. TP in Nansha wetland showed a decreasing trend from 2018 to 2019, and the TP maximum occurred in January of 2019.

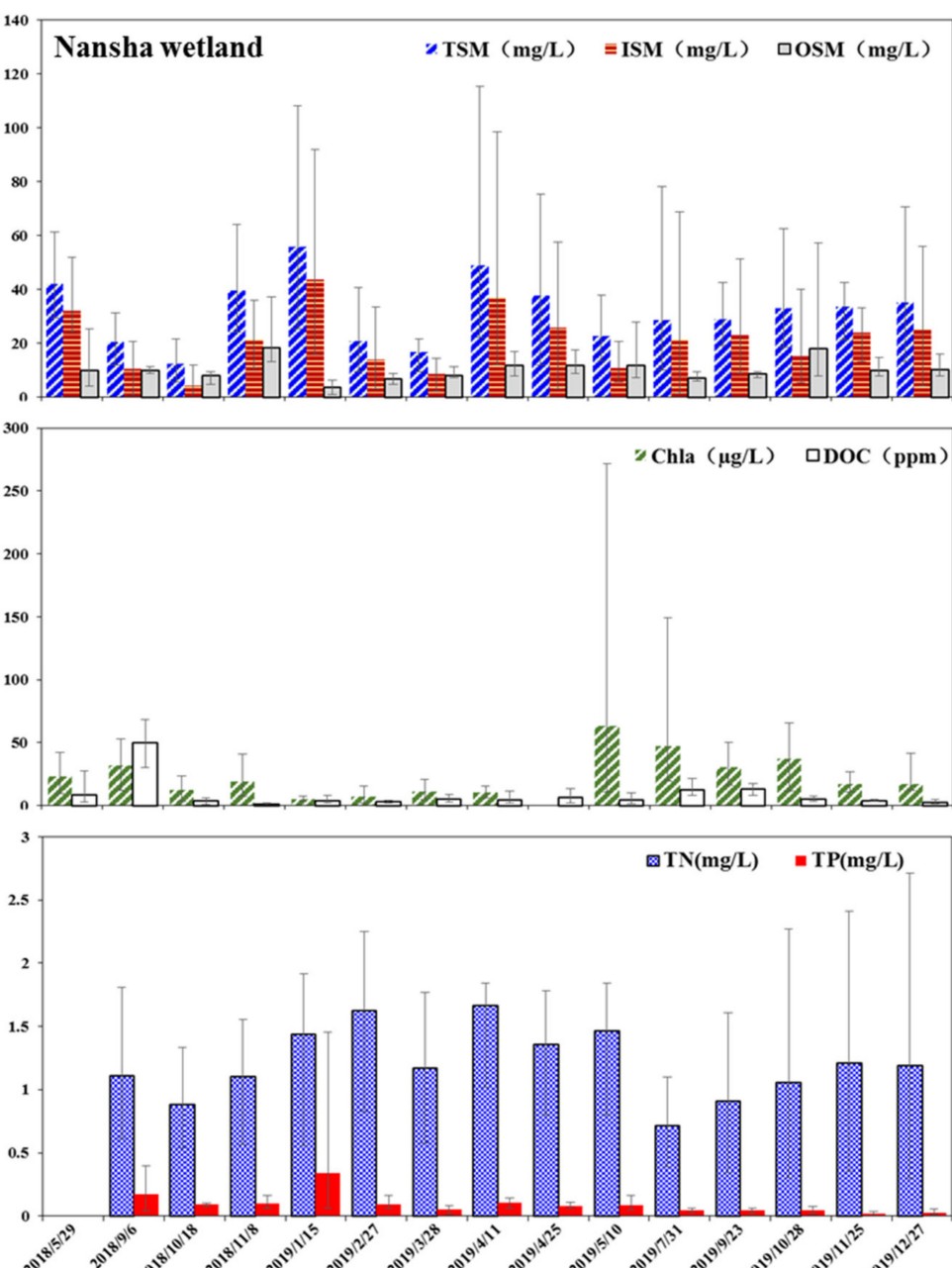

**Figure 5.** Temporal variations of water quality indicators in Nansha wetland.

## 4. Discussion

### *4.1. Water Quality Comparison of Different Wetland Types*

#### 4.1.1. Water Quality Differences between Tianfu and Nansha Wetlands

Based on data acquired by citizen scientists and NIGLAS researchers, there was a significant difference in water quality between the wetlands. Tianfu wetland, a river wetland located in the lower reaches of Lake Taihu Basin [21], was relatively clear and in a mesotrophic state [20]. Nansha wetland, a typical coastal river wetland, is influenced by tidal variations of Lingdingyang Ocean [22] and was highly turbid and in a eutrophic state.

#### 4.1.2. Water Quality Comparison between Different Wetland Uses

There were clear differences in water quality related to wetland use (Figure 6). Water quality in the natural wetlands was much better than that of the other two types. Spearman analysis on wetland types and water quality for Tianfu (Table 4) indicated that there were

significant differences between wetland types, especially for OSM and nutrients. OSM and nutrients in natural wetland were the lowest, and those of the agricultural wetland were the highest. For Nansha, the nutrient concentrations of the agricultural wetland were significantly higher than that of the wetland park and natural wetland, consistent with the published research [23]. Spearman analysis also indicated that OSM, DOC and TN are sensitive to different wetland uses. The concentration of organic matter in natural wetland water was significantly higher than that in other types of wetlands, while that in agricultural wetland was relatively low. The TN of natural wetland water was lower, while that of agricultural wetland was the highest.

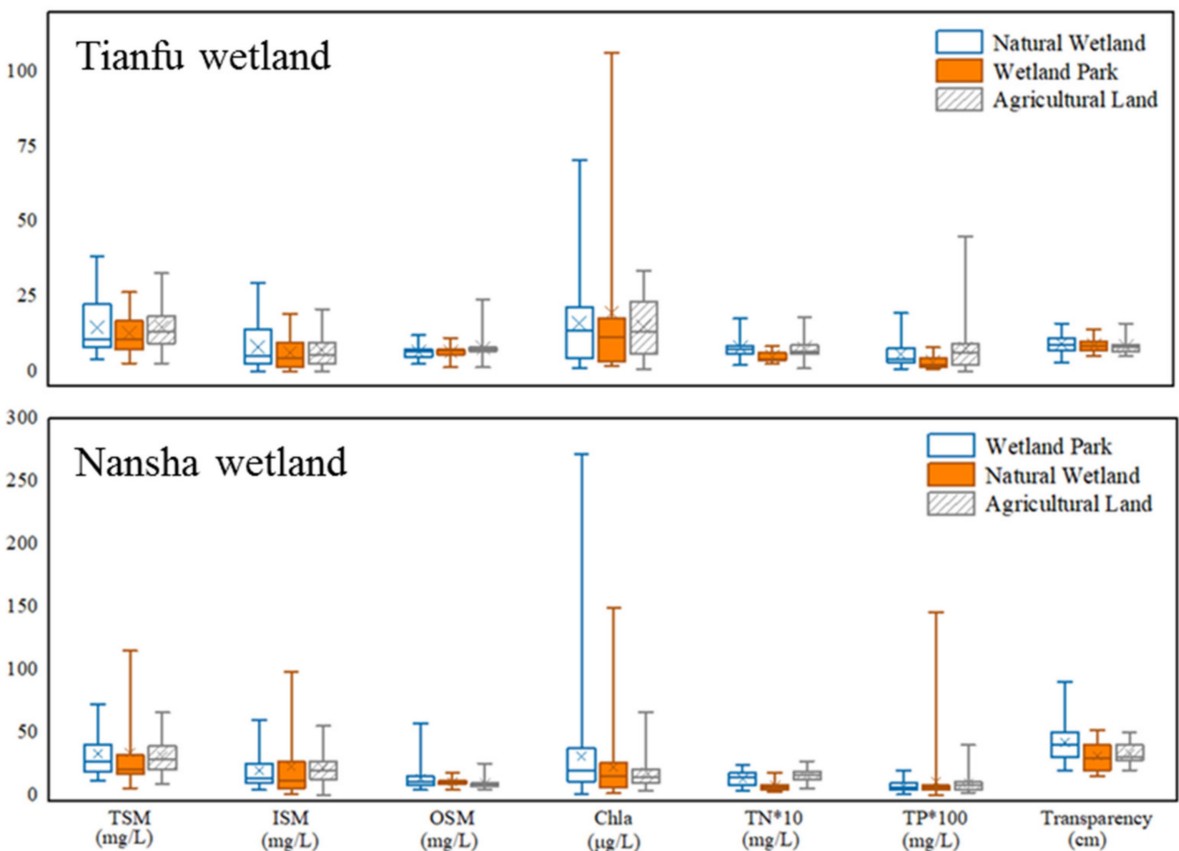

**Figure 6.** Water quality comparison of different type wetlands in Tianfu and Nansha wetlands.

**Table 4.** Spearman analysis on wetland types and water quality.

| Spearman Analysis | | Water Quality Indicators | | | | | | | |
|---|---|---|---|---|---|---|---|---|---|
| | | Secchi | TSM | ISM | OSM | Chl*a* | DOC | TN | TP |
| Tianfu wetland | r | −0.186 | 0.115 | 0.073 | 0.229 | 0.116 | 0.143 | 0.377 | 0.377 |
| | Sig. (2-tailed) | 0.104 | 0.317 | 0.525 | 0.044 | 0.311 | 0.212 | 0.001 | 0.001 |
| | N | 90 | 90 | 90 | 90 | 90 | 78 | 84 | 84 |
| Nansha wetland | r | 0.065 | 0.120 | 0.201 | −0.239 | 0.042 | −0.357 | 0.579 | 0.187 |
| | Sig. (2-tailed) | 0.588 | 0.280 | 0.068 | 0.030 | 0.719 | 0.002 | 0.000 | 0.090 |
| | N | 90 | 90 | 90 | 90 | 90 | 90 | 84 | 84 |

In this study, natural wetland, wetland park and agricultural wetland were defined as 1, 2 and 3 respectively.

### 4.1.3. Water Quality Differences between Upstream and Downstream

T-tests were used to analyze the difference between incoming and outgoing waters and the different wetland types. For Nansha, there was significant difference in the concentrations of TSM (t = 3.099, $p$ = 0.004, N = 15), ISM (t = 3.037, $p$ = 0.005, N = 15) and OSM (t = 2.240, $p$ = 0.034, N = 15) between the upstream and downstream waters of the natural wetland. The concentrations of TN showed a significant difference between upstream and downstream of wetland park (t = 3.281, $p$ = 0.003, N = 14) and agricultural wetland (t = −1.717, $p$ = 0.098, N = 14). Similarly, in Tianfu, the concentrations of TN in the wetland park also showed a significant difference between upstream and downstream (t = −3.445, $p$ = 0.002, N = 14). Other water quality indicators in Tianfu did not show significant differences.

### 4.2. Main Factors of Water Quality Differences in Tianfu and Nansha Wetlands

The highly productive and complementary aspects of wetland sediment, plants and microbes play a key role in the retention of nitrogen and phosphorus, both of which drive eutrophication [24]. The processes of pollutant removal involve sedimentation, dissolution, biological adsorption, and biochemical reactions mediated by microbial nitrification, denitrification, etc. [7,25,26]. The water purifying efficiencies of natural wetlands are highly variable based on wetland characteristics, which include area, spatial structure, location, loading and retention time, precipitation and temperature [24,27].

### 4.2.1. Water Environment and Hydrological Characteristics of Study Wetlands

Despite low water velocity in Tianfu, both wetlands exchanged water with rivers and the surrounding environment. Field activities were concentrated around the wetland; however, data from the local environmental protection department showed that the water quality of more than 75% of rivers in the Tianfu wetland area were better than Level III. For Nansha, although the water quality of Jiaomen waterway and Hongliqi waterway met water quality of level II, the water quality of the rivers connected with the waterway was poor, generally level IV or lower. These data confirmed that the water quality measured in the two wetlands was consistent with water quality monitoring in their surrounding environment.

### 4.2.2. Climatic Characteristics

Air temperature and precipitation of Tianfu and Nansha wetlands from 2018 to 2019 were shown in Figure 7. Higher temperature favors phytoplankton growth [28] and microbial decomposition of organic matter, when the nutrient and DOC conditions permit. In summer, the chlorophyll-a and DOC concentrations in Nansha were 366% and 300% higher than that in winter. However, for Tianfu wetland, only DOC had similar changes, while chlorophyll-a concentrations were relatively low and stable throughout the year. Temperature will also influence the absorption and transformation capacity of wetlands, their vegetation and microbial community [24,29,30]. In summer, the nitrogen removal efficiency of wetlands is expected to increase, largely due to denitrification processes favored by higher temperatures [31–33]. TN concentrations in the two wetlands were lower in summer and autumn than the winter and spring. This was particularly evident in the Nansha wetland. Precipitation will change the water retention and pollution loads [34,35], increasing runoff from surfaces in the catchment with consequent sediment and nutrients inflows. TSM did not show a significant correlation with precipitation in two wetlands ($p$ > 0.05, N = 15). ISM, OSM and other water quality indicators did not show any relationship to precipitation in either wetland.

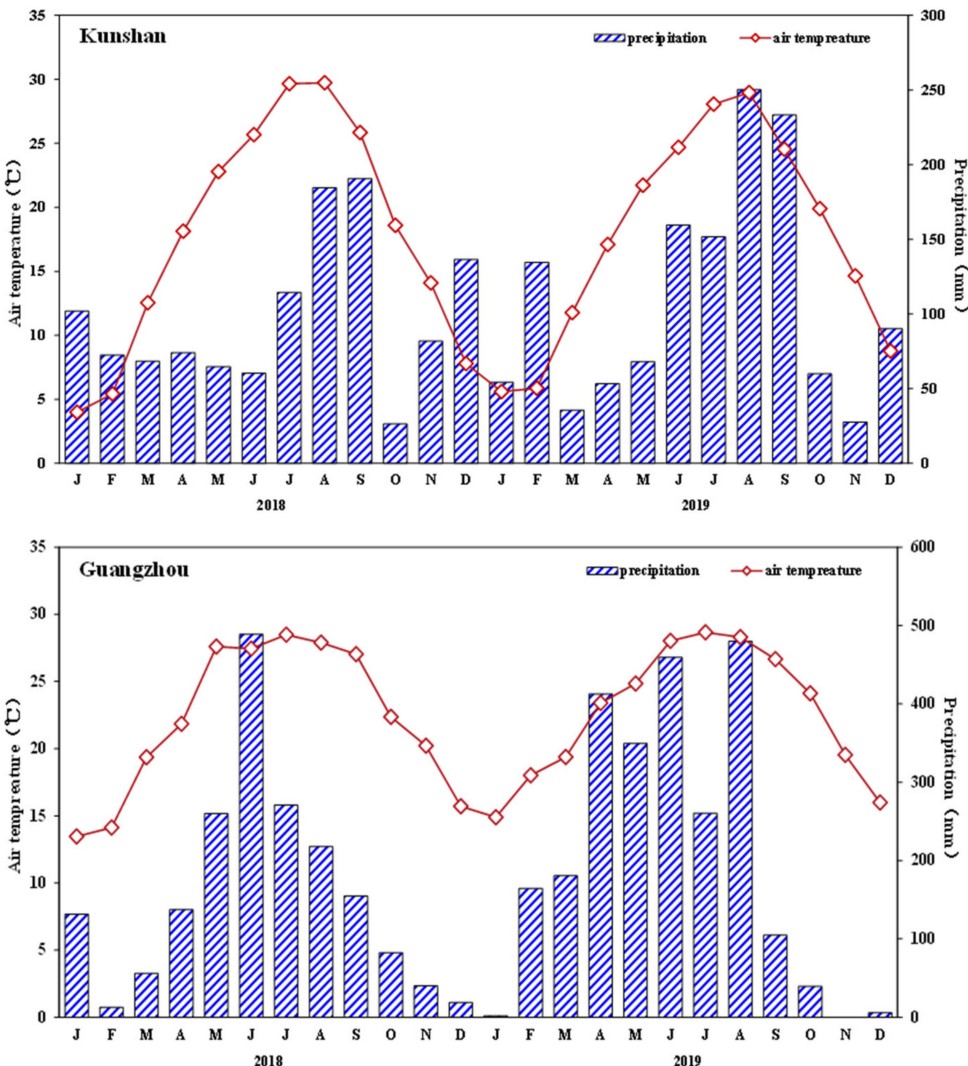

**Figure 7.** Air temperature and precipitation of Tianfu and Nansha wetlands from 2018 to 2019.

### 4.2.3. Human Activities

The agricultural area of the Tianfu wetland is used mostly for growing rice and rapeseed, using a double cropping approach. The paddy field discharges water in April and August every year, which leads reduced water quality of the agricultural wetlands. The agricultural wetlands in Nansha Wetland mainly grow cash crops (such as lotus root and banana etc.) and fish. Both agricultural wetlands presented much poorer water quality and a reduced nutrient and sediment reduction capacity with respect to the other wetland uses. For Nansha, the natural wetland was particularly effective in reducing TSM, ISM and OSM.

## 5. Conclusions

Based on the data gathered by citizen scientists in the Nansha and Tianfu wetlands, it was possible to identify different levels of ecosystem service provision between three land uses in these multiuse wetlands. Our results indicated that wetland water quality is closely related to the external water environment, with the water quality of the Tianfu wetland being much better than that of Nansha.

Our results also show that areas of the wetland used for agriculture provided the lowest water quality and the lowest level of supporting ecosystem services. In Nansha, which is challenged by the poor water quality of its surrounding catchment, the capacity of the wetland to retain and transform suspended solids and nitrogen was greatly reduced in

the part of the wetland dedicated to agricultural activities. This loss in ecosystem services has knock-on impacts on the water quality delivered to the local rivers and coastal waters.

The impact on supporting ecosystem services was less evident in the Tianfu wetland, which has a better baseline water quality in the surrounding waters. The difference between wetland uses was less evident, as were the differences in water quality-related ecosystem services. The potential to retain some areas of the wetland for sustainable agricultural uses is less impactful.

The study also demonstrated the benefits of citizen scientist-based measurements to monitor wetland conditions and inform wetland managers. Nutrient concentrations made in the field clearly showed the differences between wetlands and wetland uses. Likewise, differences in Secchi measurements reflected measured differences in suspended matter measured in the laboratory. Finally, the novel filter and filtrate color analysis approach helped to validate the increased organic particulate matter in Nansha wetland. This increased spatial and temporal coverage provided important information in the present analysis, while the wetland vegetation surveys made by citizen scientists provided important classification and validation for remote sensing-based land use/cover analysis. The advantages of involving citizen scientists to increase data availability and awareness need to be considered within the context of a more participatory management of the wetlands, and the associated costs of training these participants with low-cost standard methods.

Sustainable wetland management is a major challenge where urbanization and development pressures push for the conversion of these areas. The development of approaches to quantify and monitor the multiple ecosystem services provided by these ecosystems allows for a stronger, evidence-based decisions by wetland managers, provincial and national decision makers. By training and equipping citizen scientists to contribute to their monitoring, better tools to manage these critical ecosystems are made available and an increased distributed awareness of their importance is achieved.

**Author Contributions:** Conceptualization, Y.Z. (Yuchao Zhang) and S.L.; methodology, Y.Z. (Yuchao Zhang); investigation, M.H., Q.C. and Y.J.; writing—original draft preparation, Y.Z. (Yuchao Zhang); writing—review and editing, Y.Z. (Yuchao Zhang) and S.L.; project administration, Y.Z. (Yimo Zhang), Q.W. and X.S. All authors have read and agreed to the published version of the manuscript.

**Funding:** Funding was provided by HSBC under the HSBC Sustainable Leadership Programme through Earthwatch Europe.

**Institutional Review Board Statement:** Not applicable.

**Informed Consent Statement:** Not applicable.

**Data Availability Statement:** The raw data supporting the conclusions of this article will be made available by the authors, without undue reservation.

**Acknowledgments:** We gratefully acknowledge the citizen scientists who participated in the Sustainable Training Programme (STP) and Sustainable Leadership Programme (SLP) events and collected data. All of the satellite data and meteorological data were supported by "Lake-Watershed Science Data Center, National Earth System Science Data Sharing Infrastructure, National Science & Technology Infrastructure of China (http://lake.geodata.cn, (accessed on 10 January 2021))".

**Conflicts of Interest:** The authors declare no conflict of interest.

## Appendix A

**Table A1.** Summary of sampling events in 2018 & 2019.

| Locations | Date | Investigators | The Number of Sites | The Number of Citizen Scientists | The Quantity of Data |
|---|---|---|---|---|---|
| Nansha Wetland Park (Guangzhou) | 29/5/2018 | NIGLAS | 6 | - | 78 |
| | 15/1/2019 | | 6 | - | 90 |
| | 27/2/2019 | | 6 | - | 90 |
| | 31/7/2019 | | 6 | - | 90 |
| | 23/9/2019 | | 6 | - | 90 |
| | 28/10/2019 | | 6 | - | 90 |
| | 25/11/2019 | | 6 | - | 90 |
| | 27/12/2019 | | 6 | - | 90 |
| | 6~7/9/2018 | citizen scientists | 6 | 21 | 84 |
| | 18~19/10/2018 | | 6 | 21 | 90 |
| | 8~9/11/2018 | | 6 | 20 | 90 |
| | 28~29/3/2019 | | 6 | 20 | 90 |
| | 11~12/4/2019 | | 6 | 18 | 90 |
| | 25~26/4/2019 | | 6 | 19 | 90 |
| | 9~10/5/2019 | | 6 | 19 | 90 |
| Tianfu Wetland Park (Shanghai) | 3/7/2018 | NIGLAS | 6 | - | 78 |
| | 3/1/2019 | | 6 | - | 90 |
| | 25/1/2019 | | 6 | - | 90 |
| | 26/2/2019 | | 6 | - | 90 |
| | 26/3/2019 | | 6 | - | 90 |
| | 28/4/2019 | | 6 | - | 90 |
| | 29/7/2019 | | 6 | - | 90 |
| | 30/11/2019 | | 6 | - | 90 |
| | 30/12/2019 | | 6 | - | 90 |
| | 13~14/9/2018 | citizen scientists | 6 | 18 | 84 |
| | 1~2/11/2018 | | 6 | 19 | 90 |
| | 29~30/11/2018 | | 6 | 17 | 90 |
| | 23~24/5/2019 | | 6 | 20 | 90 |
| | 5~6/9/2019 | | 6 | 19 | 90 |
| | 17~18/10/2019 | | 6 | 20 | 90 |
| Total | | | | 251 | 2664 |

**Table A2.** Classification for surface water area functions due to National Environmental Quality Standard for Surface Water of China (GB 3838-2002) [20].

| Classification | The Purpose for Use and Protection Target |
|---|---|
| I | Mainly applicable for the source of water or the State Nature Reserve. |
| II | Mainly applicable for first-grade surface sources protection zones for domestic and drinking water, habitats of endangered aquatic organisms, fish and shrimp spawning grounds and feeding grounds etc. |
| III | Mainly applicable for second-grade surface sources protection zones for domestic and drinking water, fish and shrimp wintering grounds and migration channels, aquacultural grounds of fish, shrimp, shellfish and aquatic plants, swimming areas and etc. |
| IV | Mainly applicable for general industrial use and recreational water areas for human indirect contact. |
| V | Mainly applicable for agricultural and general landscape requirement use. |

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
