# Peer review of "Comparing Wetland Ecosystems Service Provision under Different Management Approaches: Two Cases Study of Tianfu Wetland and Nansha Wetland in China"

_sustainability, doi:10.3390/su13168710_

Round 1
Reviewer 1 Report
In my perspective, this article is more interesting if the authors can put more emphasis on how citizen science can work well in water quality monitoring. This can be a model that can be applied by other cases or sites.
I didn't see strong scientific findings unless the authors could provide comprehensive discussion on ecosystem services with sufficient literature review or comparison study. I'm interested in the statement that water and hydrological condition has effects on wetland capacity in maintaining water quality (see the abstract), but this manuscript didn't really show it well with a deep analysis and discussion. My suggestion is to improve the manuscript based on some comments, as follows:
Abstract:
- Please make it more concise, especially in the first part.
Introduction:
- It is worth to give a short description of terminology 'blue green infrastructure' in the beginning of introduction part.
Study area and Methodology:
- I think these two sections can be merged into a section of 'Methodology'. You can group it into: study area, citizen science methods-data, spatial analysis, lab analysis, etc..
- Section 2.2.2: does 'simple water quality measurement kit' refer to section 3.1.2 ?
- Table 1 can be put as supplementary data.
- What is the population of the study sites ?
- There are many statements that can be simplified or omitted. For example, information about Sentinel satellite operation is not necessary explained.
Results:
- This section should comprise not only tables or figures of the results but also data analysis that is mostly stated in your discussion section. In other way, you can just have one section of 'Result and Discussion'.
- From my understanding, NIGLAS researchers and citizen scientists collected samples separately. How different were the results obtained from those groups ? Do the figures and tables show only lab results from NIGLAS collection ?
- It would be better if section 4.1 paragraph 1 is described as a table.
- Figure 1 indicates delineation of study sites (Tianfu and Nansha 'wetland') but Figure 2 shows the 'wetland park', which is part of management type of the wetland. Can you clarify it ?
Discussion:
- Discussion is too short. I wonder if there is any study previously conducted in these wetlands ? This can be the supporting information.
- Section 5.2.1: Readers might not be familiar with water quality level classification. It is better to briefly explain this classification. The last statement: " this confirmed that the water quality was consistent with surrounding environment". What does it mean ?
- Section 5.2.2: Chl-a in Tianfu is relatively stable. Do you think that it is related to nutrient and DOC condition ?
Author Response
In my perspective, this article is more interesting if the authors can put more emphasis on how citizen science can work well in water quality monitoring. This can be a model that can be applied by other cases or sites.
I didn't see strong scientific findings unless the authors could provide comprehensive discussion on ecosystem services with sufficient literature review or comparison study. I'm interested in the statement that water and hydrological condition has effects on wetland capacity in maintaining water quality (see the abstract), but this manuscript didn't really show it well with a deep analysis and discussion. My suggestion is to improve the manuscript based on some comments, as follows:
Abstract:
Please make it more concise, especially in the first part.
Response: We revised the first part of abstract due to this comment.
Pls see abstract in detail.
Introduction:
It is worth to give a short description of terminology 'blue green infrastructure' in the beginning of introduction part.
Response: In the revised version, we added some short sentences about the introduction of blue green infrastructure and its benefits.
“Blue-Green Infrastructure is an approach opted by many urban cities to combat cli-mate change and environmental degradation (https://theconstructor.org/sustainability/blue-green-infrastructure/555236/). Blue-Green Infrastructure is a network of natural and semi-natural areas, offering environmental, economic, and social benefits to communities (M. Charlesworth and Warwick, 2020).”
Pls see the first paragraph of Section 1 in detail.
Study area and Methodology:
I think these two sections can be merged into a section of 'Methodology'. You can group it into: study area, citizen science methods-data, spatial analysis, lab analysis, etc..
Response: We have group the sections of Data and Methodology into a new section of “Methodology and data”.
Pls see section 2 in detail.
Section 2.2.2: does 'simple water quality measurement kit' refer to section 3.1.2 ?
Response: Yes. Citizen scientists utilized the simple water quality measurement kit to achieve some in-situ water quality indicators, such as Secchi Disc, Water Color, Turbidity, NO3--N, PO43--P.
Table 1 can be put as supplementary data.
Response: We have moved this table to appendix.
Pls see Appendix 1 in detail
What is the population of the study sites ?
Response: There are 6 fixed monitoring points in Tianfu and Nansha wetlands respectively, which were set at the upstream and down stream of natural wetland, agricultural wetland and wetland park.
Pls see Figure 1 and Section 2.1 in detail.
There are many statements that can be simplified or omitted. For example, information about Sentinel satellite operation is not necessary explained.
Response: We have simplified the introduction of Sentinel satellite operation.
Pls see Section 2.3 in detail.
Results:
This section should comprise not only tables or figures of the results but also data analysis that is mostly stated in your discussion section. In other way, you can just have one section of 'Result and Discussion'.
Response: Thank a lot for this comments. All authors think that the section of Results just states the water quality conditions, but the section of discussion focuses on the comparison of water quality in different wetland types and locations. So we didn’t put these two sections into one.
From my understanding, NIGLAS researchers and citizen scientists collected samples separately. How different were the results obtained from those groups ? Do the figures and tables show only lab results from NIGLAS collection ?
Response: When citizen science training activity was held, NIGLAS researchers and citizen scientists collected samples together. If there was no citizen science training activity, NIGLAS collected sample at the same sampling points once a month. Citizen scientists used simple measurement kit to achieve qualitative or classified water quality data. NIGLAS took collected samples to the lab and used specific methods to determine exact and continuous water quality data.
It would be better if section 4.1 paragraph 1 is described as a table.
Response: In the revised version, we added a new table to describe land cover use in Tianfu and Nansha Wetlands.
Pls see section 3.1 and Table 2 in detail.
Figure 1 indicates delineation of study sites (Tianfu and Nansha 'wetland') but Figure 2 shows the 'wetland park', which is part of management type of the wetland. Can you clarify it ?
Response: Thanks a lot for this comments. We clarify these word in the revised version. Tianfu Wetland Park includes three different wetlands: natural wetland, agricultural wetland and wetland park. But there are only natural wetland and wetland park located in Nansha Wetland Park. The agricultural wetland doesn’t belong to Nansha Wetland Park. So they are called Nansha wetland together.
Pls see the caption of Figure 1 and Figure 2.
Discussion:
Discussion is too short. I wonder if there is any study previously conducted in these wetlands? This can be the supporting information.
Response: Thanks for this comment. There are few published literatures on water quality of these two wetlands. We had added a Chinese literature on water quality variation of Nansha wetland.
Pls see Section 4.1.2 in detail.
Section 5.2.1: Readers might not be familiar with water quality level classification. It is better to briefly explain this classification.
Response: According to the National Environmental Quality Standard for Surface Water of China (GB 3838-2002), water of rivers and lakes has been classified into five levels due to water quality (Appendix 2).
Appendix 2 Classification for surface water area functions due to National Environmental Quality Standard for Surface Water of China (GB 3838-2002) (SEPA 2002)
|
Classification |
The purpose for use and protection target |
|
I |
Mainly applicable for the source of water or the State Nature Reserve. |
|
II |
Mainly applicable for first-grade surface sources protection zones for domestic and drinking water, habitats of endangered aquatic organisms, fish and shrimp spawning grounds and feeding grounds etc. |
|
III |
Mainly applicable for second-grade surface sources protection zones for domestic and drinking water, fish and shrimp wintering grounds and migration channels, aquacultural grounds of fish, shrimp, shellfish and aquatic plants, swimming areas and etc. |
|
IV |
Mainly applicable for general industrial use and recreational water areas for human indirect contact. |
|
V |
Mainly applicable for agricultural and general landscape requirement use. |
Pls see Appendix 2 indetail.
The last statement: " this confirmed that the water quality was consistent with surrounding environment". What does it mean ?
Response: Because both of these two wetlands exchanged water with rivers and surrounding environment, surrounding environment affected the water quality of wetland significantly. For example, the water quality of Tianfu wetland was much better than that of Nansha wetland , because the water quality of more than 75% of rivers in the Tianfu wetland area were better than Level III.
Pls see section 4.2.1 in detail.
Section 5.2.2: Chl-a in Tianfu is relatively stable. Do you think that it is related to nutrient and DOC condition ?
Response: Pearson correlation analysis showed that there are not any correlations between Chla concentrations and nutrients/DOC concentrations in Tianfu Wetland. The detail data is shown as followed.
|
Water quality indicator |
Pearson analysis |
DOC |
TN |
TP |
|
Chla |
r |
-.004 |
.166 |
-.138 |
|
Sig. |
.979 |
.294 |
.382 |
|
|
N |
45 |
42 |
42 |
Reviewer 2 Report
The conceptual discussion should be more in-depth and with more international bibliographic references making the comparison with another Studies in China. Examples:
- Wetlands are the most common natural blue green infrastructure and provide well defined ecosystem ser-vices (Costanza et al. 1997), including: 1) food products; 2) flood prevention/mitigation; 3) improvement of water quality and microclimate; 4) conserve the biodiversity; and 5) social/cultural functions. There have been many studies exploring the mechanisms related to the improvement of surface runoff water quality in wetlands, such as nitrate loss by denitrification (Bachand and Horne 2000) and phosphorus removal by soil absorption (Gray et al. 2000a).
- Few studies have compared the relative benefits on water quality mitigation related to different wetland uses.
To sustain, from a theoretical point of view, the technical results presented, using more specialized literature.
Advantages and disadvantages of the proposed method, compared with results obtained in other studies.
In the conclusion, include difficulties and disadvantages of the methodology and techniques used, as well as the challenges and lines of future research.
Author Response
The conceptual discussion should be more in-depth and with more international bibliographic references making the comparison with another Studies in China. Examples:
- Wetlands are the most common natural blue green infrastructure and provide well defined ecosystem services (Costanza et al. 1997), including: 1) food products; 2) flood prevention/mitigation; 3) improvement of water quality and microclimate; 4) conserve the biodiversity; and 5) social/cultural functions. There have been many studies exploring the mechanisms related to the improvement of surface runoff water quality in wetlands, such as nitrate loss by denitrification (Bachand and Horne 2000) and phosphorus removal by soil absorption (Gray et al. 2000a).
- Few studies have compared the relative benefits on water quality mitigation related to different wetland uses.
Response: Thanks for this comment. We have added some new international references into these parts above-mentioned. Almost all the literatures are about the impact of a special land use type on water quality change. Only one Chinese paper has done a similar study on Nansha wetland, but the research data is only limited to winter and summer of a year, not continuous monitoring results. But we have similar conclusions. We have added this literature into the discussion section.
Pls see Sections 1 and 4.1.2.
To sustain, from a theoretical point of view, the technical results presented, using more specialized literature.
Response: Thanks for this comment. We have added some more specialized literatures into the method and discussion sections.
Pls see Sections 2.2.1 and 4.2.
Advantages and disadvantages of the proposed method, compared with results obtained in other studies.
Response: Thanks for this comment. There are few published literatures on water quality of these two wetlands. We had added a Chinese literature on water quality variation of Nansha wetland.
Pls see Section 4.1.2 in detail.
In the conclusion, include difficulties and disadvantages of the methodology and techniques used, as well as the challenges and lines of future research.
Response: Thanks for this comment. In the revised version, we added some sentences on difficulties and disadvantages of involving citizen scientists to increase data availability and awareness, as well as the challenges of future research.
Pls see the Section 5.
Reviewer 3 Report
reconsider the citations according to Instructions for Authors
"in the text In the text, reference numbers should be placed in square brackets [ ], and placed before the punctuation; for example [1], [1–3] or [1,3]. For embedded citations in the text with pagination, use both parentheses and brackets to indicate the reference number and page numbers; for example [5] (p. 10). or [6] (pp. 101–105)".
reconsider the References according to Instructions for Authors
Page 2
denitrification(Bachand and Horne 2000) – consider change to denitrification [3]
The annual average temperature is 15.7 ℃- consider change to the annual average temperature is 15.7°C
Page 3
temperature is 21.9 ℃ - consider change to temperature is 21.9°C
Page 6
Samples (2L) – consider change to Samples (2 L)
dried at 105 °C – consider change to dried at 105°C
at 450 °C for – consider change to at 450°C for
the filters(Cao et al. 2020; Ma et al. 2006) – consider change to the filters [number of reference].
UV2600 spectrophotometer - consider change to UV2600 spectrophotometer (manufacturer and country)
Page 10
in P-PO4 concentrations – consider change to in P-PO4 concentrations
TN concentrations were high throughout the year, f higher in winter and spring, and lower in summer and autumn – consider change to TN concentrations were high throughout the year, higher in winter and spring, and lower in summer and autumn
Author Response
reconsider the citations according to Instructions for Authors
"in the text In the text, reference numbers should be placed in square brackets [ ], and placed before the punctuation; for example [1], [1–3] or [1,3]. For embedded citations in the text with pagination, use both parentheses and brackets to indicate the reference number and page numbers; for example [5] (p. 10). or [6] (pp. 101–105)".
reconsider the References according to Instructions for Authors
Response: In the revised version, we have revised the citation and references according to the Instructions for Authors.
Page 2
denitrification(Bachand and Horne 2000) – consider change to denitrification [3]
Response: In the revised version, we have changed the citations according to the Instructions for Authors.
Pls see the first paragraph of Section 1 in detail.
The annual average temperature is 15.7 ℃- consider change to the annual average temperature is 15.7°C
Response: We have revised it.
Pls see the third paragraph of Section 2.1 in detail.
Page 3
temperature is 21.9 ℃ - consider change to temperature is 21.9°C
Response: We have revised it.
Pls see the fourth paragraph of Section 2.1 in detail.
Page 6
Samples (2L) – consider change to Samples (2 L)
Response: We have revised it.
Pls see the first paragraph of Section 2.4 in detail.
dried at 105 °C – consider change to dried at 105°C
Response: We have revised it.
Pls see the second paragraph of Section 2.4 in detail.
at 450 °C for – consider change to at 450°C for
Response: We have revised it.
Pls see the second paragraph of Section 2.4 in detail.
the filters(Cao et al. 2020; Ma et al. 2006) – consider change to the filters [number of reference].
Response: In the revised version, we have changed the citations according to the Instructions for Authors.
Pls see the second paragraph of Section 2.4 in detail.
UV2600 spectrophotometer - consider change to UV2600 spectrophotometer (manufacturer and country)
Response: We have added these information into the revised version.
Pls see the second paragraph of Section 2.4 in detail.
Page 10
in P-PO4 concentrations – consider change to in P-PO4 concentrations
Response: We have revised to “P-PO4 concentrations”.
Pls see the second paragraph of Section 3.3.1 in detail.
TN concentrations were high throughout the year, f higher in winter and spring, and lower in summer and autumn – consider change to TN concentrations were high throughout the year, higher in winter and spring, and lower in summer and autumn
Response: We have deleted “f” in the sentence.
Pls see the first paragraph of Section 3.3.2 in detail.
Round 2
Reviewer 1 Report
The authors have addressed most of my comments.
Please do the spelling checks and consistency (eg. and - &) for minor correction.
Author Response
he authors have addressed most of my comments.
Please do the spelling checks and consistency (eg. and - &) for minor correction.
Response: Thanks a lot for the reviewer’s comments. In the revised version, the authors have checked the whole manuscript, and have made the revisions including text, tables and figures. All revisions have been marked with red color.